# Generating Synthetic Datasets for Few-shot Prompt Tuning

**Xu Guo,*, Zilin Du, Boyang Li & Chunyan Miao**
Nanyang Technological University, Singapore

## Abstract

A major limitation of prompt tuning is its dependence on large labeled training datasets. Under few-shot learning settings, prompt tuning lags far behind full-model fine-tuning, limiting its scope of application. In this paper, we leverage the powerful LLMs to synthesize task-specific labeled data for training the soft prompts. We first introduce a distribution-aligned weighted generator tuning (DawGen) method to encourage generating in-distribution data that aligns with the few-shot real data. Then, we train soft prompts on both synthetic and real datasets using a gradient surgery approach, which eliminates the conflicting gradients from different data sources. Experiments on seven sentence-pair classification datasets demonstrate the effectiveness of our proposed method for boosting prompt tuning in few-shot learning settings. Results on QQP, MRPC, and SICK datasets are even comparable to the performance of transfer learning from large real-world datasets, showing the promise of synthetic data as an alternative for enhancing soft prompt tuning.

## 1 Introduction

As Large Language Models (LLMs) increase in size, adapting them to downstream tasks by fine-tuning (FT) a separate copy for each task becomes unfeasible. Prompt Tuning (Lester et al., 2021) (PT) emerges as a solution to this challenge by freezing the LLM and instead training a set of soft prompts pre-pended to the input data in an end-to-end manner. Compared with other parameter-efficient learning methods such as adapter tuning (Houlsby et al., 2019) and LoRA (Hu et al., 2022), PT makes no changes to the model architecture and can be applied to a frozen model with a static computational graph, enabling fast and flexible deployment. On a wide range of downstream tasks, PT has shown comparable performance as FT (Lester et al., 2021; Liu et al., 2022). However, recent studies indicate that PT requires sufficient labeled training data to achieve competitive performance as FT, yet in few-shot settings, PT significantly underperforms FT (Gu et al., 2022; Guo et al., 2022).

To boost PT in few-shot learning tasks, previous methods mainly focus on finding a better initialization for soft prompts (Gu et al., 2022; Guo et al., 2022). This is achieved by pre-training the soft prompts on a large-scale real-world corpus such as OpenWebText (Gokaslan & Cohen, 2019a) or similar source-domain datasets (Gu et al., 2022; Vu et al., 2022; Guo et al., 2022). However, these approaches bear a common limitation: the dependence on large real-world datasets. On the one hand, online text corpora often exhibit substantial domain discrepancies varying across different downstream tasks. On the other hand, source-domain datasets are often not readily available, particularly for low-resource and emerging domains.

Recently, there has been a growing interest in generating training data with LLMs (Ye et al., 2022a; Meng et al., 2022; 2023; Yu et al., 2024). This paper extends this line of research to tackle the limitation of prompt tuning in few-shot learning settings, aiming to bypass the need for large-scale labeled training data. Specifically, we employ a source LLM to generate a synthetic training set for the task at hand, which can be treated as a medium for carrying the pre-learned knowledge from a source LLM, to train soft prompts for a target LLM to achieve enhanced few-shot learning performance.

---

*Please contact Xu Guo (xu008@e.ntu.edu.sg) for future questions.

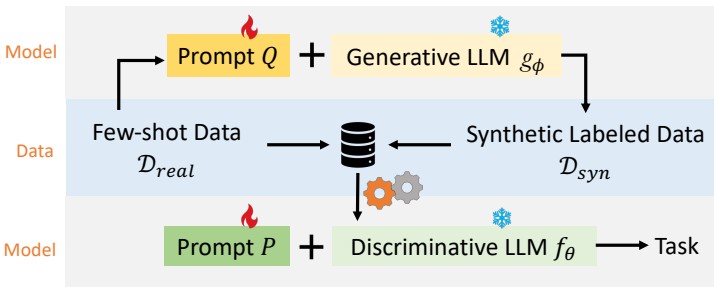

**Step 1**: traing a soft prompt $Q$ for a frozen LLM $g_\phi$ on the few-shot training set $\mathcal{D}_{real}$. **Step 2**: apply $g_\phi$ and $Q$ to generate a synthetic labeled training set $\mathcal{D}_{syn}$. **Step 3**: With $\mathcal{D}_{real}$ and $\mathcal{D}_{syn}$, train a soft prompt $P$ for a discriminative LLM $f_\theta$ to perform downstream tasks, while tackling issues like gradient conflicts.

Figure 1: A schematic overview of the framework.

We show an overview diagram of our framework in Figure 1. We divide the learning procedure into three steps. First, to encourage the source LLM $g_\phi$ to generate in-domain data, we train label-conditional soft prompts $Q$ using a few real-world samples, $\mathcal{D}_{real}$, to adapt the frozen $g_\phi$ to the task domain. However, given the limited number of real-world samples, $Q$ can easily overfit shortcut tokens (Du et al., 2021; Tang et al., 2023) and mislead $g_\phi$ to generate texts that are not relevant to the task label, resulting in a poor distribution for $\mathcal{D}_{syn}$. Hence, we introduce Distribution-Aligned Weighted GENerator tuning (DawGen[1]) which encourages the generated texts under the same class to be semantically close while those under different classes to be semantically distant. Second, we apply the adapted generator $g_{\phi,Q}$ to generate a synthetic training set, $\mathcal{D}_{syn}$, for the task at hand. However, directly combining synthetic data with the few-shot real data for training can lead to an interference effect, due to their potential distribution gap. To better exploit the synthetic data while making the best use of few-shot real data, we employ gradient surgery (Yu et al., 2020) to modify the gradients computed on the synthetic data by subtracting the components that oppose the direction of gradients from the real data. The trained soft prompts followed by an input example are used to prompt the target LLM for label prediction. They can also be combined with hard prompts to confer dual benefits (Gao et al., 2021; Gu et al., 2022).

Extensive experiments on seven sentence-pair classification tasks and two LLM backbones demonstrate the effectiveness of our proposed framework for enhancing PT in few-shot settings. Notably, PT with 102K parameters outperforms FT with 770M parameters by a large margin in few-shot settings and can even achieve comparable performance to transfer learning using extensive real-world datasets on QQP, MRPC, and SICK.

## 2 Related Work

### 2.1 Few-shot Learning with Pre-trained Language Models

Fine-tuning pre-trained language models has been the standard practice in few-shot learning, where a language model and a task-specific head are tuned together for a given task (Zhang et al., 2021; Gao et al., 2021; Liu et al., 2022; Zhang et al., 2022). However, fine-tuning the entire model on a few training samples (e.g., 16 samples per class) often leads to overfitting. One possible remedy is to manually craft hard prompts, consisting of natural language instructions and demonstrations (Brown et al., 2020; Mishra et al., 2022) for LLMs to perform in-context learning without updating any model parameters. Nevertheless, their effectiveness greatly relies on the skills of the prompt engineer and the prompts they write.

Instead of manually crafted hard prompts, prompt tuning (Lester et al., 2021; Zhang et al., 2022) learns soft prompt vectors from training data and PT on T5-large can outperform the results of manual prompting on GPT-3. Still, PT performance greatly depends on the availability of substantial training data (Gu et al., 2022; Guo et al., 2022). In few-shot settings, PT significantly underperforms FT. To address this limitation, researchers have studied utilizing online corpora to pre-train soft prompts under self-supervised learning objectives (Gu et al., 2022), such as next sentence prediction (Devlin et al., 2019), or applying

---

[1]https://github.com/guoxuxu/soft-prompt-transfer/tree/main/DawGen

transfer learning from datasets in similar domains (Guo et al., 2022). This paper provides an alternative solution by employing a more powerful LLM to generate synthetic training data for training soft prompts, thus circumventing the demand for extensive real-world data.

## 2.2 LLMs as Task-specific Training Data Generators

Early efforts in generating synthetic data with language models were to augment the existing dataset with generic texts (Kumar et al., 2020; Puri et al., 2020; Anaby-Tavor et al., 2020). E.g., Puri et al. (2020) use GPT-2 to generate a synthetic corpus to substitute the Wikipedia corpus to improve the performance of QA tasks. These texts are not tailored to any downstream task and therefore can not be used to train task-specific models. As LLMs continue to grow in size, recent works have shifted towards creating a new paradigm for playing with LLMs, i.e., distilling task-specific training data directly from a frozen LLM to enhance downstream tasks (Schick & Schütze, 2021b; Ye et al., 2022a; Meng et al., 2022; Gao et al., 2023; Meng et al., 2023; Yu et al., 2024). These synthetic datasets are then used to boost downstream models such as DistillBERT and RoBERTa (Meng et al., 2022; 2023).

However, when prompted with a simple label-conditional natural language prompt, the LLM generator can easily forget label information when generating long sequences (Li et al., 2022; Zhong et al., 2024) and therefore may generate samples that are not associated with the given labels. Moreover, the LLM generator, when frozen for inference, tends to generate texts that follow its pertaining data distribution, which often exhibits a domain gap with the task at hand (Guo & Yu, 2022). This paper provides a distribution-aligned weighted generator tuning method to mitigate these issues.

## 2.3 Learning with Synthetic Training Data

The existence of low-quality samples can be detrimental to model training. Synthetic datasets are often generated at scale, which is inevitable to contain low-quality data (Gao et al., 2023). Recent works on synthetic text generation with LLMs (Ye et al., 2022a; Meng et al., 2022; 2023) adopt various training strategies to exploit the synthetic training data. For example, ZeroGen (Ye et al., 2022a; Gao et al., 2022) employs a noise-robust loss function (Ghosh et al., 2017) for learning with synthetic training data. FewGen (Meng et al., 2023) adopts label smoothing and temporal ensembling (Laine & Aila, 2016) to degrade the confidence level of model prediction during training. ProGen (Ye et al., 2022b) incorporates a quality estimation module to select the synthetic dataset. These methods are primarily employed to work with synthetic data; however, in our paper, where we use a few real samples for supervision, they cannot address the disparity between real and synthetic data. Hence, we propose to use the gradient surgery method (Yu et al., 2020) to directly alter the conflicting gradients on the synthetic data, thereby enhancing the learning performance.

## 3 Synthesizing Training Data for Prompt Tuning

### 3.1 Preliminaries

**Prompt Tuning**. Lester et al. (2021) converts all downstream tasks into a text-to-text generation format (Raffel et al., 2020) in order to reduce the gap between pre-training and downstream tasks. Taking sentence-pair classification as an example, given a labeled training example $(X, y) \in \mathcal{D}, |\mathcal{D}| = N$, where $X = [S_1, S_2]$ represents a sentence pair and $y \in \mathcal{Y}$ denotes a class label. For example, in paraphrase detection, the label space $\mathcal{Y}$ may include two labels, yes and no, indicating if two sentences are paraphrases. In order to fully utilize the pretrained model, we devise a task-specific natural language prompt $H$ and a template $\mathcal{T}(\cdot)$ that reformat the original task as a cloze-style task. $\mathcal{T}(X) = \{H, X, [\text{MASK}]\}$. One example for paraphase detection is "Are $\langle S_1 \rangle$ and $\langle S_2 \rangle$ equivalent? [MASK]". An LLM, $f_\theta$, predicts the label $y$ at the position of the [MASK] token.

Prompt tuning is done by prepending $n$ tokens with trainable prompt embeddings, $P \in \mathbb{R}^{n \times d}$, to the template $\mathcal{T}$. Throughout the training, $\theta$ remains unchanged and $P$ is optimized

by minimizing the cross-entropy loss on the training set $\mathcal{D}$:

$$\mathcal{L}_{ce}(P) = -\mathbb{E}_{(X,y)\in\mathcal{D}} \log \text{Pr}_\theta([\text{MASK}] = y|[P; E]) \tag{1}$$

**Synthetic Text Generation.** Taking sentence-pair classification as an example. Given the label space $\mathcal{Y} = \{Y_l\}_{l=1}^L$, we compose a label-conditional prompt for each $Y_l$ using the template $\mathcal{T}_c(Y_l) = \{H, Y_l, X\}$, e.g., "Sentence 1 and sentence 2 are equivalent. Sentence 1: $\langle S_1 \rangle$. Sentence 2: ". Then, an autoregressive LLM, $g_\phi$, takes $\mathcal{T}_c(Y_l)$ as the context and generates a subsequent sequence of tokens $x_{1:K}$ that maximize the joint likelihood:

$$\prod_{j=1}^K \text{Pr}_\phi(x_j|x_{<j}; \mathcal{T}_c(Y_l)). \tag{2}$$

The decoding process stops when the end-of-sentence token is predicted or the maximum sequence length, $K$, is reached. The generated sentence for $S_2$ is expected to satisfy the relationship $Y_l$ with $S_1$. To encourage generating diverse $X_{syn}$ for the same label $Y_l$, we employ a stochastic decoding algorithm (e.g., top-k and nucleus sampling). The generated sentence pairs and the given label $y_l$ establish a synthetic dataset $\mathcal{D}_{syn} = \{(X_{syn}, y_{syn})\}$.

### 3.2 Distribution-Aligned Weighted Generator Tuning

For every downstream task, we use the few-shot real dataset, $\mathcal{D}_{real}$, to perform domain adaptation for the generator $g_\phi$ to improve the quality of the synthetic dataset $\mathcal{D}_{syn}$.

**Parameter-Efficient Generator Tuning.** Recent LLMs for text generation often have billions of parameters. As such, tuning the entire model $\phi$ for domain adaptation is impractical. Parameter-efficient methods, such as prompt tuning (Lester et al., 2021) and prefix tuning (Li & Liang, 2021), arise as an alternative to full-model fine-tuning by pre-pending a few external prompt embeddings, $Q$, to the input (or every transformer layer's output as done by prefix tuning), and training $Q$ on the domain-specific data while keeping $\phi$ unchanged.

For a downstream task with $L$ classes, we can train one soft prompt $Q_l \in \mathbb{R}^{n \times d}$ for each class label $Y_l$. The training objective for tuning $Q_l$ in this setting is the standard language modeling objective. The generator parameters $\phi$ are frozen and only the soft prompt $Q_l$ is optimized on the real training set $\mathcal{D}_{real}$:

$$\mathcal{L}_{gen}(Q_l) = -\frac{1}{|\mathcal{D}_{real}|} \sum_{X\in\mathcal{D}_{real}, y=Y_l} \sum_{x_j\in X} \log \text{Pr}_\phi(x_j|x_{<j}; Q_l). \tag{3}$$

**Weighted Generator Tuning.** The above language modeling objective treats all tokens equally. To encourage the data generator network to generate label-discriminative texts, FewGen (Meng et al., 2023) propose to train a weight net, $\Phi_W : \mathbb{R}^d \mapsto \mathbb{R}$, which learns to assign higher weights to those generated tokens that are more likely to discriminate the ground-truth label $Y_l$ from other labels $Y_{l'} \in \mathcal{Y}$ by minimizing the weighted generation loss,

$$\mathcal{L}_{wGen}(Q_l) = -\mathbb{E}_{X\in\mathcal{D}_{real}, Y=Y_l} \mathbb{E}_{x_j\in X} W_j \cdot \log \text{Pr}_\phi(x_j|x_{<j}; Q_l). \tag{4}$$

Here, $Q$ and $W$ are optimized under the bi-level optimization framework. That is, first optimizing the loss $\mathcal{L}_{wGen}$ produces a function $Q$ of $W$: $Q(W)$, which is then applied a second time to optimize the weight net, $W$, by minimizing the following loss:

$$\mathcal{L}_{disc}(W) = -\mathbb{E}_{x_j\in X} \frac{\text{Pr}_\phi(x_j|x_{<j}; Q_l(W))}{\sum_{l'} \text{Pr}_\phi(x_j|, x_{<j}; Q_{l'}(W))}. \tag{5}$$

By optimizing $Q$ and $W$ iteratively, the generator $g_{\phi,Q}$ learns to generate tokens that are more related to the given label than other labels.

**Algorithm 1: Generator Tuning.**

**Data:** Few-shot real dataset $\mathcal{D}_{real}$.
**Initialize:** Prompts $Q$, Pre-trained LLM $g_\phi$.

1   initialize $t = 0$;
2   **while** $t < T + 1$ **do**
3     $t += 1$;
4     $\mathcal{B} \leftarrow$ Sample a batch from $\mathcal{D}_{real}$;
5     $Q^t(W^t) \leftarrow$ Take a gradient descent step on $\mathcal{B}$ with $\mathcal{L}_{\text{DawGen}}(Q^t; W^t)$;
6     $W^{t+1} \leftarrow$ Take a gradient descent step on $\mathcal{B}$ with $\mathcal{L}_{\text{disc}}(Q^t(W^t))$;
7     $Q^{t+1} \leftarrow$ Take a gradient descent step on $\mathcal{B}$ with $\mathcal{L}_{\text{DawGen}}(Q^t; W^{t+1})$;
8   **end**
**Output:** Generator $g_{\phi,Q}$

**Algorithm 2: Prompt Tuning.**

**Data:** Few-shot $\mathcal{D}_{real}$ and synthetic $\mathcal{D}_{syn}$.
**Initialize:** Prompts $P$, Pre-trained LLM $f_\theta$.

1   initialize $t = 0$;
2   **while** $t < T + 1$ **do**
3     $t += 1$;
4     $\mathcal{B}_{real}, \mathcal{B}_{syn} \leftarrow$ Sample a batch from $\mathcal{D}_{real}, \mathcal{D}_{syn}$;
5     Compute gradients $\delta_{real} = \frac{\partial \mathcal{L}_{ce}(P)}{\partial P}$ on $\mathcal{B}_{real}$;
6     Compute gradients $\delta_{syn} = \frac{\partial \mathcal{L}_{ce}(P)}{\partial P}$ on $\mathcal{B}_{syn}$;
7     **if** $\delta_{syn} \cdot \delta_{real} < 0$ **then**
8       $\delta'_{syn} = \delta_{syn} - \text{Proj}_{\delta_{real}}(\delta_{syn})$;
9     **end**
10    $\delta = \delta_{real} + \epsilon \cdot \delta'_{syn}$;
11    $P \leftarrow P - \eta \cdot \delta$;
12   **end**
**Output:** Soft prompt $P$

**Distribution-Aligned Regularization.** However, given the few-shot training set, it is easy for the weight net to overfit shortcut tokens, which are not robust tokens for the given task. For example, the token "not" can discriminate the positive sentence "It is a good movie" from the negative sentence "It is not a good movie", but is obviously not a generalizable label-discriminative token. Hence, enforcing the generator to solely rely on the weights may lead to generating sentences that are irrelevant to the given label. We propose to regularize the generator tuning objective by adding a sentence-level distribution constraint to encourage the generated sentence to align with the in-distribution data:

$$\mathcal{L}_{\text{DawGen}}(Q) = \mathbb{E}_{l \in [1,L]} \mathcal{L}_{\text{wGen}}(Q_l) + \mathcal{L}_{\text{dist}}(Q), \tag{6}$$

where

$$\mathcal{L}_{\text{dist}}(Q) = \mathbb{E}_{(X,y) \in \mathcal{D}_{real}} max(0, 1 - D(W \cdot Z_{i,l}, W \cdot Z_{j,l}) + D(W \cdot Z_{i,l}, W \cdot Z_{j,l'})). \tag{7}$$

Here, $W \in \mathbb{R}^{1 \times K}$ indicates the weights for a sequence of $K$ tokens. $Z_{i,l} = g_{\phi,Q}(X_i), Z_{i,l} \in \mathbb{R}^{K \times d}$, denotes the last-layer hidden states output from the generator $g_{\phi,Q}(\cdot)$ for $i$-th instance $X_i$ of class $Y_l$, and $Z_{j,l'}$ represents that from a different class $Y_{l'}$. $D(\cdot, \cdot)$ measures the cosine similarity between two vectors. By minimizing $\mathcal{L}_{\text{dist}}(Q)$, we encourage the generated texts to stay close to the ones in the same class while being pulled away from those that belong to other classes. We present the whole procedures in Algorithm 1.

### 3.3   Training Soft Prompts with Synthetic Data Augmentation

Despite the domain adaptation procedure employed, the resulting synthetic training set can inevitably contain low-quality data. Training soft prompts with a naive combination of synthetic and few-shot real data can result in the optimization process being dominated by gradients from the synthetic data. Hence, we up-sample the few-shot data by pairing each batch of synthetic data with a corresponding batch from the few-shot real data. Then, we employ gradient surgery to these paired batches to resolve conflicting gradients from different data sources in prompt tuning.

**Gradient Surgery.** It was first proposed to de-conflict gradients in a multi-task learning setting, where a model $\theta$ is trained on a set of $M$ tasks (Yu et al., 2020). Let $\delta_i = \frac{\partial \mathcal{L}_i(\theta)}{\partial \theta}$ denote the gradients of $i$-th task loss $\mathcal{L}_i(\theta)$ with respect to the model $\theta$. $\forall \delta_i \in \{\delta_i\}_{i=1}^M$, $\delta_i$ is iteratively altered across all the other tasks by subtracting the component $\frac{\delta_i \cdot \delta_j}{\|\delta_j\|^2} \delta_j$, which is its projection to the plane of $j$-th task' gradient $\delta_j$, where $j \neq i$. This step is applied when $\delta_i \cdot \delta_j < 0$, which indicates the two tasks have interference in driving the optimization path.

In this paper, we treat the gradients from real data, $\delta_{real}$, as the positive gradients for the task and always project the gradients of the synthetic data, $\delta_{syn}$, to the direction of $\delta_{real}$:

$$\text{Proj}_{\delta_{real}}(\delta_{syn}) = \frac{\delta_{syn} \cdot \delta_{real}}{\delta_{real} \cdot \delta_{real}} \delta_{real} = \frac{\delta_{syn} \cdot \delta_{real}}{\parallel \delta_{real} \parallel} \cdot \frac{\delta_{real}}{\parallel \delta_{real} \parallel}, \tag{8}$$

where $\frac{\delta_{real}}{\parallel\delta_{real}\parallel}$ is the normal plane of the gradients $\delta_{real}$ from real data, and $\frac{\delta_{syn}\cdot\delta_{real}}{\parallel\delta_{real}\parallel}$ is the magnitude of the projection of $\delta_{syn}$ onto this normal plane. If $\delta_{syn} \cdot \delta_{real} < 0$, then the projected gradients will be dropped and the gradients of synthetic data will be modified as:

$$\delta'_{syn} = \delta_{syn} - \text{Proj}_{\delta_{real}}(\delta_{syn}). \tag{9}$$

By removing the conflicting gradients of the synthetic data, we train soft prompts using the loss function in Equation 1 and update the weights with a gradient descent approach:

$$P \leftarrow P - \eta(\delta_{real} + \epsilon \cdot \delta'_{syn}), \tag{10}$$

where $\eta$ is the learning rate and $\epsilon$ is a factor for controlling the strength of synthetic knowledge guidance, which was studied in a similar work in computer vision (Zhu et al., 2023). The whole algorithm is presented in Algorithm 2.

## 4 Experiments

### 4.1 Datasets, Metrics, and Settings

We conduct evaluations on seven sentence-pair classification datasets in two tasks. In the paraphrase detection task, we use MRPC (Dolan & Brockett, 2005) and QQP[2]. In the natural language inference task, we use MNLI (Williams et al., 2018), SNLI (Bowman et al., 2015), QNLI (Rajpurkar et al., 2016), RTE (Dagan et al., 2005), and SICK (Marelli et al., 2014). We follow LM-BFF (Gao et al., 2021) to prepare the few-shot learning setting: both $\mathcal{D}_{\text{train}}$ and $\mathcal{D}_{\text{dev}}$ contain 16 samples per class, which are sampled from the original training set using 5 random seeds, and the original development set is used as the test set. We adopt Accuracy for all the classification tasks and report the average test accuracy over 5 seeds. We compare methods using the average performance across the seven datasets. More details about the datasets, models, and training settings can be found in the Appendix.

### 4.2 Baselines

We consider the following zero-shot and few-shot baseline methods. We also compared with two transfer learning methods, **SPOT** (Vu et al., 2022) and **OPTIMA** (Guo et al., 2022), where a large-scale real-world source-domain dataset is available.

**(Zero-shot) Prompting.** We prompt the frozen T5-large and Flan-T5-large with only task-specific natural language prompts (i.e., hard prompts) and treat it as the zero-shot learning baseline. For fair comparisons, we apply the same hard prompts for all baselines where applicable. Prompt-based templates are presented in the Appendix.

**In-context Learning (ICL).** Following GPT-3 (Brown et al., 2020), we incorporate the acquired few-shot examples as demonstrations in the hard prompt templates, which is also called few-shot prompting (in contrast to zero-shot prompting). The role of few-shot examples helps the frozen T5-large and Flan-T5-large models better understand the task by exemplifying the task instruction with real-world examples. The order of few-shot samples is randomly determined and we report the average performance across five runs.

**Full-model Fine-tuning (FT).** We feed the few-shot data without any hard prompts into T5-large and Flan-T5-large and fine-tune the entire networks. Different from traditional fine-tuning that trains an additional classification layer from scratch, here, we apply the label verbalizer and tune the language modeling head instead.

---

[2]https://quoradata.quora.com/First-Quora-Dataset-Release-Question-Pairs

| Method | #Trainable Params | QQP | MRPC | MNLI | SNLI | QNLI | RTE | SICK | AVG |
|---|---|---|---|---|---|---|---|---|---|
| | | | | | T5-large | | | | |
| Prompting | 0 | 42.63 | 33.80 | 33.20 | 33.31 | 49.46 | 52.35 | 14.51 | 37.04 |
| In-Context | | 59.55 | 33.52 | 34.49 | 33.80 | 49.72 | 48.52 | 40.90 | 42.93 |
| FT | | **72.50** | 61.72 | 42.82 | 48.90 | 50.11 | 55.81 | **77.90** | 58.54 |
| Prompt-based FT | 770M | 60.15 | 59.66 | 42.94 | **54.16** | 51.75 | 57.18 | 69.98 | 56.55 |
| PFT + soft prompt | | 60.22 | 56.18 | 43.86 | 48.45 | 57.38 | 55.60 | 76.23 | 56.85 |
| PPT | 410K | 46.11 | 52.37 | 34.05 | 35.28 | 52.86 | 48.59 | 45.64 | 44.99 |
| Prompt Tuning | 102K | 47.28 | 58.94 | 33.29 | 33.21 | 52.68 | 51.70 | 27.80 | 43.49 |
| Ours | 102K | 66.77 | **69.67** | **53.20** | 46.81 | **69.84** | **57.40** | 72.73 | **62.35** |
| SPOT[†] | 102K | 64.5 | 68.7 | 74.3 | 78.8 | - | - | 72.9 | - |
| OPTIMA[†] | | 69.1 | 71.2 | 78.4 | 82.1 | - | - | 73.3 | - |
| | | | | | Flan-T5-large | | | | |
| Prompting | 0 | 62.15 | 67.71 | 62.13 | 64.07 | 80.29 | 26.35 | 33.31 | 56.57 |
| In-Context | | **82.84** | 75.27 | 62.44 | 54.87 | **89.98** | 19.06 | 38.02 | 60.35 |
| FT | | 79.17 | 78.29 | 79.76 | 86.37 | 56.86 | **86.57** | **83.73** | 78.68 |
| Prompt-based FT | 770M | 80.28 | 78.04 | 78.42 | **88.11** | 50.56 | 84.84 | 80.96 | 77.32 |
| PFT + soft prompt | | 79.64 | 77.65 | **79.87** | 86.90 | 80.37 | 84.91 | 70.60 | **79.99** |
| Prompt Tuning | 102K | 70.40 | 72.82 | 59.89 | 63.26 | 83.73 | 26.78 | 60.61 | 62.49 |
| Ours | | 82.14 | **78.40** | 71.84 | 82.43 | 88.80 | 56.82 | 79.88 | 77.19 |

Table 1: Test accuracy on all datasets. The best result of each dataset is bolded. [†]: SPOT and OPTIMA are transfer learning techniques that require a fully labeled source dataset, with performance numbers from (Guo et al., 2022)

**Prompt-based Full-model Fine-tuning (PFT).** Using hard prompts for model tuning is represented by LM-BFF (Gao et al., 2021) and PET (Schick & Schütze, 2021a). Here, we apply the same hard prompts as other baselines to wrap every training sample for tuning T5-large and Flan-T5-large. We also consider incorporating soft prompts for model tuning as P-Tuning (Liu et al., 2022) and DART (Zhang et al., 2022), denoted as *PFT + soft prompt*.

**Pre-trained Prompt Tuning (PPT).** Proposed by Gu et al. (2022) to pre-train soft prompts on text corpus from OpenWebText (Gokaslan & Cohen, 2019b) using the next sentence prediction objective (Devlin et al., 2019). We download the pre-trained checkpoint for T5-xxl and fine-tune them on the sentence-pair classification tasks.

**FewGen.** Proposed by Meng et al. (2023), where authors first use the few-shot data to adapt CTRL (Keskar et al., 2019) to every task with prefix tuning and then generate synthetic datasets using the adapted generator. We use their code[3] to produce the synthetic datasets (denoted as FewGen) for all tasks and train soft prompts for T5-large or Flan-T5-large. It can be treated as the ablation for DawGen where only $\mathcal{L}_{\text{wGen}}$ is applied.

### 4.3 Few-shot learning performance

Our main results are presented in Table 1. Overall, compared with the naive prompt tuning method, our approach yields an average improvement of approximately 18% across all datasets when applied to T5-large, and about 15% for Flan-T5-large. In particular, **PT** (using 102K parameters) under our framework outperforms **FT** (using 770M parameters) by an average improvement of 3.8% across all datasets based on T5-large. When compared to SPOT and OPTIMA, which use large real-world datasets for transfer learning, our approach exhibits competitive performance on QQP, MRPC, and SICK, though its performance on

---

[3]https://github.com/yumeng5/FewGen

| Method | Generator | QQP | MRPC | MNLI | SNLI | QNLI | RTE | SICK | AVG |
|--------|-----------|-----|------|------|------|------|-----|------|-----|
| T5-Large | | | | | | | | | |
| Real+Syn | | 52.64 | **70.53** | 38.15 | 33.96 | 57.08 | 52.64 | 48.01 | 50.43 |
| Real+Syn+LS | | 53.09 | 67.94 | 38.58 | 34.11 | 56.99 | 56.03 | 58.05 | 52.11 |
| Real → Syn | FewGen | 59.51 | 70.04 | 47.46 | 41.99 | **65.52** | **57.91** | 65.81 | 58.32 |
| Syn → Real | | 63.78 | 68.92 | 36.97 | 35.17 | 63.24 | 53.86 | 52.90 | 53.55 |
| (Real, Syn) | | **66.44** | 68.12 | **48.03** | **44.81** | 64.15 | 56.54 | **68.46** | **59.51** |
| Real+Syn | | **62.86** | **70.38** | 43.97 | 35.62 | 60.11 | 53.29 | 51.31 | 53.93 |
| Real → Syn | DawGen | 62.60 | 69.39 | 47.94 | 46.14 | 66.33 | **58.12** | 61.48 | 58.85 |
| Syn → Real | | 62.69 | 69.28 | 42.45 | 38.01 | 60.35 | 55.31 | 56.20 | 54.89 |
| (Real, Syn) | | 61.77 | 69.99 | **48.76** | **45.10** | **66.37** | 57.20 | **70.80** | **59.99** |
| Flan-T5-large | | | | | | | | | |
| Real+Syn | | 81.13 | 76.82 | 67.91 | 66.79 | 85.32 | 54.01 | 75.04 | 72.43 |
| Real+Syn+LS | | 79.56 | 76.06 | **73.50** | 71.42 | 82.96 | 55.88 | 70.37 | 72.82 |
| Real → Syn | FewGen | 79.09 | 76.08 | 64.94 | 63.46 | 85.76 | 57.19 | 71.15 | 71.10 |
| Syn → Real | | 82.18 | **79.00** | 68.35 | 72.24 | 82.08 | **58.70** | 77.88 | 74.34 |
| (Real, Syn) | | **82.33** | 78.04 | 68.86 | **80.14** | **87.19** | 56.68 | **78.56** | **75.97** |
| Real+Syn | | 83.60 | 76.81 | 71.85 | 72.48 | 84.11 | 53.72 | 69.17 | 73.10 |
| Real → Syn | DawGen | 80.50 | 75.64 | 66.42 | 69.41 | 86.52 | **54.22** | 73.33 | 72.29 |
| Syn → Real | | **83.26** | **78.55** | **72.15** | 77.29 | **87.51** | 50.76 | 72.17 | 74.53 |
| (Real, Syn) | | 81.83 | 76.96 | 70.18 | **79.69** | 87.38 | 51.63 | **76.97** | **74.94** |

Table 2: Test accuracy of different strategies for learning on real and synthetic data, where synthetic datasets are generated by FewGen and DawGen respectively. The best result in each group is bolded.

MNLI and SNLI remains a big challenge, highlighting the possibility of using synthetic training data as an alternative to enhance few-shot prompt tuning.

When comparing Prompt Tuning (PT) with both Prompting and In-Context Learning baselines, we observed that PT's performance enhancement is marginal, despite leveraging 102K parameters to learn from data. This suggests that PT can be prone to overfitting when applied to few-shot datasets. On the contrary, FT employs 770M parameters and, despite its greater tendency to overfit the data, surpasses PT by an average of 14-18%, indicating that a better pre-trained initialization can significantly enhance the learning outcomes.

## 4.4 The order in which synthetic and real data appears does matter

Previous research (Vu et al., 2022; Gu et al., 2022; Guo et al., 2022) indicates that data-driven initialization significantly improves PT. Here, we study how the order in which synthetic data and real data appear affects PT. We experimented with several strategies: **Real+Syn** directly combines the synthetic and few-shot real data as a new training set and performs shuffling before mini-batch training. **Real → Syn** trains soft prompt first on $\mathcal{D}_{real}$ and then on $\mathcal{D}_{syn}$ in every training epoch. **Syn → Real**, on the contrary, trains soft prompt first on $\mathcal{D}_{syn}$ and then on $\mathcal{D}_{real}$ in every training epoch. **(Real, Syn)** means pairing every batch sampled from $\mathcal{D}_{syn}$ with a batch sampled from $\mathcal{D}_{real}$ and combining them as a new batch to train soft prompts, which is employed in our approach. Results are presented in Table 2.

We observed that a naive combination of **Syn + Real** data performs the worst, where the few-shot real data could be overwhelmed by the larger synthetic data. Simply applying a label smoothing regularization, denoted as **Syn + Real + LS**, generally does not help. In contrast, **(Real, Syn)**, which up-samples the few-shot real data when paired with either FewGen or DawGen data, confers an obvious improvement for both T5-large and Flan-T5-large. We also observed an interesting phenomenon, where T5-large prefers **Real → Syn** while Flan-T5-large prefers **Syn → Real**. This may suggest that training a model initially on the few-shot real data is not always an advantage and the order in which real and synthetic data are presented impacts differently for different LLM backbones.

| Generator | Real | GS | QQP | MRPC | MNLI | SNLI | QNLI | RTE | SICK | AVG |
|---|---|---|---|---|---|---|---|---|---|---|
| | | | T5-Large | | | | | | | |
| FewGen | | | 56.70 | 69.25 | 42.18 | 34.63 | 56.91 | 53.87 | 34.11 | 49.66 |
| | ✓ | | 66.44 | 68.12 | 48.03 | 44.81 | 64.15 | 56.54 | 68.46 | 59.51 |
| | ✓ | ✓ | 67.85 | 70.05 | 49.52 | 46.39 | 66.96 | 55.96 | 72.08 | 61.25 |
| DawGen | | | 58.62 | 69.11 | 44.30 | 36.63 | 61.97 | 55.16 | 50.39 | 53.74 |
| | ✓ | | 61.77 | 69.99 | 48.76 | 45.10 | 66.37 | 57.20 | 70.80 | 59.99 |
| | ✓ | ✓ | 66.77 | 69.67 | 53.20 | 46.81 | 69.84 | 57.40 | 72.73 | 62.35 |
| | | | Flan-T5-large | | | | | | | |
| FewGen | | | 78.29 | 78.72 | 62.03 | 66.13 | 84.00 | 50.54 | 69.81 | 69.93 |
| | ✓ | | 82.33 | 78.04 | 68.86 | 80.14 | 87.19 | 56.68 | 78.56 | 75.97 |
| | ✓ | ✓ | 81.76 | 78.36 | 75.26 | 81.07 | 87.67 | 61.13 | 79.26 | 77.78 |
| DawGen | | | 83.11 | 77.05 | 68.21 | 70.49 | 84.07 | 49.02 | 68.66 | 71.52 |
| | ✓ | | 81.83 | 76.96 | 70.18 | 79.69 | 87.38 | 51.63 | 76.97 | 74.94 |
| | ✓ | ✓ | 82.14 | 78.40 | 71.84 | 82.43 | 88.80 | 56.82 | 79.88 | 77.19 |

Table 3: Test accuracy of ablation studies. "GS" stands for "Gradient Surgery" in prompt tuning.

## 4.5 Ablation study

We evaluate every component in our approach and present the results in Table 3. Specifically, we observe that: **1)** the distribution-aligned regularization term $\mathcal{L}_{\text{dist}}$ is effective - Comparing **DawGen** against **FewGen**, the soft prompts trained exclusively on DawGen results in an average improvement of 4% for T5-large and 1.6% for Flan-T5-large across all datasets compared to the one trained on FewGen; **2)** utilizing a few real samples to augment synthetic datasets is beneficial, and this benefit is more pronounced on FewGen than DawGen, indicating that the higher the quality of a synthetic dataset, the less it requires supplementation with real data supervision. **3)** the gradient surgery technique effectively mitigates the conflict between synthetic and real data sources - applying gradient surgery further improves the performance by an average of approximately 2% across the datasets.

## 4.6 Instruction-tuned models are better few-shot learners

Recent works (Varia et al., 2023; Aly et al., 2023) suggested that the advantage of instruction tuning (Ouyang et al., 2022; Chung et al., 2022) for language models can extend to few-shot learning settings. This is further corroborated in our experiments. Across the tables, Flan-T5-large excels T5-large on all experiments by a large margin. In this paper, we provide a few new findings to this avenue: **1)** instruction-tuned models can follow soft prompts better, as shown by **PFT + soft prompt** versus **PFT**, where an average improvement of 2% is spotted on Flan-T5-large while no increase is observed on T5-large. **2) Prompt Tuning** on top of instruction-tuned models are less prone to overfitting the few-shot data. It is shown that Flan-T5-large elevates the PT performance of T5-large by around 20% on average. **3)** instruction-tuned models prefer using synthetic data to warm up the learning, as shown in Table 2. Moreover, when using Flan-T5-large as the backbone, **Syn → Real** strategy leads to an average improvement of about $3 \sim 4\%$, as shown by comparing either **FewGen** or **DawGen** from Table 3 with **Syn → Real** from Table 2. While synthetic data is often used as a form of regularization for learning on real datasets, in our study, presenting a few real samples after training on synthetic data produces a regularization effect. We suspect that the instruction-following capability of Flan-T5-large may play an important role.

## 5 Conclusion

This paper presents a framework for generating synthetic training data with LLMs to boost prompt tuning in few-shot settings. We introduce Distribution-Aligned Weighted GENerator

tuning (DawGen) to encourage the generation of label-relevant text samples to improve the quality of the synthetic dataset. To better exploit the synthetic data while making the best use of the few-shot real data set, we employ the gradient surgery technique during prompt tuning to eliminate the conflicting gradients from the synthetic data batches. Experiments on seven sentence-pair classification datasets and two LLM backbones demonstrated the effectiveness of our method for boosting prompt tuning in few-shot settings with synthetic data. We show that Prompt Tuning augmented with DawGen can surpass Full-Model Fine-Tuning in few-shot settings by a large margin, highlighting the feasibility of using LLM-generated data as an alternative.

## 6 Limitations and Discussions

Currently, there are a few limitations of this study that may limit the impact scope of the insights conveyed by this paper.

- The gap between the evaluation metric and the quality of the synthetic data. Specifically, the downstream few-shot learning performance of prompt tuning, which is often measured by accuracy, may only reflect the preferences of the model rather than aligning with human judgment in terms of the quality of the synthetic data. There should be intuitive methods to explain to humans why one synthetic sample is superior to another.

- The few-shot learning performance on these public benchmarks reported in this study does not stand for the state-of-the-art. They only reflect the performance of prompt tuning. More advanced parameter-efficient learning methods like LoRA could bring higher performance than prompt tuning. Nevertheless, the relevant improvements in this study can still support the effectiveness of a specific strategy.

- Synthetic data generation cost can be a concern. Current data generators rely on large language models, which have a deep stack of transformer layers. Feedforward computations and autoregressive generations involve non-trivial GPU, memory, and time costs. A few research efforts, such as FastGen Ge et al. (2024), have been devoted to accelerating LLM inference costs.

## 7 Future Directions

In Table 4, we compare the modern augmentation paradigm, which leverages Generative AI, specifically LLMs, with the traditional paradigm that relies on transfer learning. Together with the discussions in the Limitation section, we propose the following directions: 1) employ explainable approaches to highlight the influential elements in the synthetic data and then devise quantitative measures to assess the data quality; and 2) develop inference acceleration algorithms tailored specifically for batch generation.

| Paradigms | Data Source | Label | Pretrain | Challenge |
|---|---|---|---|---|
| Transfer Learning | real-world | human annotation | ✓ | distribution gap |
| Generative AI | LLM-generated | given as prompts | × | data quality, task relevance |

Table 4: Two paradigms for boosting few-shot learning performance of soft prompt tuning.

**Acknowledgments**

This research is supported by the Wallenberg-NTU Presidential Postdoctoral Fellowship, the Nanyang Associate Professorship, and NRF Fellowship (NRF-NRFF13-2021-0006). Any opinions, findings, conclusions, or recommendations expressed in this material are those of the authors and do not reflect the views of the funding agencies.

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

## A Experimental Setup Details

### A.1 Datasets

The dataset statistics are presented in Table 5. We sample the few-shot $|\mathcal{D}_{train}|$ and $|\mathcal{D}_{dev}|$ from their original training sets and report performance on their original development sets. For MNLI, we report test accuracy on the matched version. We covert their original labels using the verbalizer as in the Table.

| Task | Train | Dev | $n_{class}$ | Verbalizers |
|------|-------|-----|---------|-------------|
| QQP | 363,846 | 40,430 | 2 | yes/no |
| MRPC | 3,068 | 408 | 2 | yes/no |
| MNLI | 392,702 | 9,815 | 3 | yes/maybe/no |
| SNLI | 549,367 | 9,842 | 3 | yes/maybe/no |
| QNLI | 104,743 | 5,463 | 2 | yes/no |
| RTE | 2,490 | 278 | 2 | yes/no |
| SICK | 4,439 | 4,906 | 3 | yes/maybe/no |

Table 5: The original dataset statistics.

### A.2 Hard prompt template.

| Task | Template | label |
|------|----------|-------|
| QQP | Question 1 is ⟨different⟩ from Question 2. Question 1: ⟨$S_1$⟩ Question 2: | different |
| | Question 1 is ⟨equivalent⟩ to Question 2. Question 1: ⟨$S_1$⟩ Question 2: | equivalent |
| MRPC | Sentence 1 is ⟨different⟩ from Sentence 2. Sentence 1: ⟨$S_1$⟩ Sentence 2: | different |
| | Sentence 1 is ⟨equivalent⟩ to Sentence 2. Sentence 1: ⟨$S_1$⟩ Sentence 2: | equivalent |
| MNLI | Sentence 1 ⟨implies⟩ Sentence 2. Sentence 1: ⟨$S_1$⟩ Sentence 2: | implies |
| | Sentence 2 ⟨supplements⟩ Sentence 1. Sentence 1: ⟨$S_1$⟩ Sentence 2: | supplements |
| | Sentence 2 ⟨contradicts⟩Sentence 1. Sentence 1: ⟨$S_1$⟩ Sentence 2: | contradicts |
| SNLI | Sentence 1 ⟨implies⟩ Sentence 2. Sentence 1: ⟨$S_1$⟩ Sentence 2: | implies |
| | Sentence 2 ⟨supplements⟩Sentence 1. Sentence 1: ⟨$S_1$⟩ Sentence 2: | supplements |
| | Sentence 2 ⟨contradicts⟩ Sentence 1. Sentence 1: ⟨$S_1$⟩ Sentence 2: | contradicts |
| QNLI | Paragraph is ⟨relevant⟩ to Question. Question: ⟨$S_1$⟩ Paragraph: | relevant |
| | Paragraph is ⟨irrelevant⟩ to Question. Question: ⟨$S_1$⟩ Paragraph: | irrelevant |
| RTE | Sentence 1 ⟨implies⟩ Sentence 2. Sentence 1: ⟨$S_1$⟩ Sentence 2: | implies |
| | Sentence 2 ⟨supplements⟩ Sentence 1. Sentence 1: ⟨$S_1$⟩ Sentence 2: | supplements |
| SICK | Sentence 1 ⟨implies⟩ Sentence 2. Sentence 1: ⟨$S_1$⟩ Sentence 2: | implies |
| | Sentence 2 ⟨supplements⟩ Sentence 1. Sentence 1: ⟨$S_1$⟩ Sentence 2: | supplements |
| | Sentence 2 ⟨contradicts⟩ Sentence 1. Sentence 1: ⟨$S_1$⟩ Sentence 2: | contradicts |

Table 6: Natural language prompt templates used for generator tuning.

### A.3 Models and Training Setting

**Generator Tuning.** Following FewGen (Meng et al., 2023), we use CTRL (1.6B parameters) (Keskar et al., 2019) as the generator $g_\phi$ and apply prefix tuning to train it on the few-shot real dataset via training a set of label-specific $Q = \{Q_l\}$ using the few-shot data $\mathcal{D}_{real}$ following Algorithm 1. We generate 1000 synthetic samples per class for every task using the same setting as FewGen[4]. Following Li & Liang (2021), we set the prefix length to the number of tokens in the natural prompts in Table 5. We use the same hyperparameters for all tasks and set the batch size to 2. The learning rate for weight net is set to 1e-2 and the one for $Q$ is set to 5e-3. The number of training epochs is fixed to 20. More details can be found in the original paper (Meng et al., 2023).

---

[4]https://github.com/yumeng5/FewGen

| Task | Template | Verbalizers |
|------|----------|-------------|
| QQP | Are ⟨S1⟩ and ⟨S2⟩ equivalent? [MASK] | yes/no |
| MRPC | Are the first sentence: ⟨S1⟩ and the second sentence: ⟨S2⟩ equivalent? [MASK] | yes/no |
| MNLI | hypothesis: ⟨S2⟩ premise: ⟨S1⟩ answer: [MASK] | yes/maybe/no |
| SNLI | hypothesis: ⟨S2⟩ premise: ⟨S1⟩ answer: [MASK] | yes/maybe/no |
| QNLI | Is the question: ⟨S2⟩ relevant to the paragraph: ⟨S1⟩ ? [MASK] | yes/no |
| RTE | hypothesis: ⟨S2⟩ premise: ⟨S1⟩ answer: [MASK] | yes/no |
| SICK | hypothesis: ⟨S2⟩ premise: ⟨S1⟩ answer: [MASK] | yes/maybe/no |

Table 7: Natural language prompt templates used for prompt tuning.

| | QQP | MRPC | MNLI | SNLI | QNLI | RTE | SICK |
|---|---|---|---|---|---|---|---|
| T5-large | - | 0.014 | 8e-3 | - | 8e-3 | 9e-4 | - |
| Flan-T5-large | 0.2 | 0.002 | - | - | 9e-8 | - | - |

Table 8: p-values for DawGen over full-model finetuning.

**Prompt Tuning.** Following previous works (Lester et al., 2021; Gu et al., 2022; Guo et al., 2022), we conduct prompt tuning on T5 (Raffel et al., 2020) and use the lm-adapted version of T5-large (770M parameters) for the main experiments. We also experimented with Flan-T5-large (Chung et al., 2022), which further trained T5-large on instruction tuning datasets to improve its instruction following capability (Ouyang et al., 2022). We fix the prompt length $n = 100$ and learning rate $\eta = 0.3$ as suggested by Lester et al. (2021). We use the cosine learning rate scheduler for all methods. We set the maximum number of training steps to 1,000 and evaluate models on the development set every 4 steps. We set batch size to 4 for MRPC and QQP, and 6 for the other NLI datasets. For all the prompt tuning experiments, the T5-large and Flan-T5-large backbone models are fixed and only soft prompt embeddings are updated. All the training experiments are done on NVIDIA A6000 with 49 GB.

## B   Significance tests

We conducted a significance test for Table 1 on results where our method was superior. Table 8 shows the p-values. Our method shows statistically significant improvement over the second-best baseline, full-model fine-tuning (FT). Although not superior in every task, our method shows higher average improvements while using only 102K parameters, compared to 770M in full-model fine-tuning.

## C   Discussion of the computational cost

Comparing the computational costs of different training paradigms directly is challenging due to variations in training data, pretraining strategies, and stopping conditions. There could be a trade-off between performance and computational cost.

- PPT: It involves a one-time pre-training of prompts on 10GB of OpenWebText data using next sentence prediction. While these prompts are reusable across tasks, they may underperform in tasks that are underrepresented in the OpenWebText corpora.

- SPoT: Prompts are pre-trained on one or more source tasks, and then adapted for a target task. However, selecting appropriate source tasks requires an extensive search and may struggle with large domain gaps.

- DawGen: It involves a one-time adaptation of the generator to the target task using few-shot examples. The adapted generator can then produce unlimited synthetic data, enhancing the flexibility of prompt training. However, the generation cost depends on the LLM backbone and the advanced inference acceleration methods such as FastGen.

