# OpenReview forum: "Generating Synthetic Datasets for Few-shot Prompt Tuning"
_colmweb.org/COLM/2024/Conference — COLM_

### Official Review · Reviewer_T1e9 · 2024-05-08

**Rating:** 6
**Confidence:** 4
**Ethics Flag:** 1

**Summary:**

This work tackles the problem that prompt tuning performs poorly in low-resource settings. It proposes a DawGen framework to generate synthetic data to augment prompt tuning and apply a gradient surgery approach to avoid conflicts from different data sources. Experiments shows that DawGen outperforms various baselines in low-resource prompt tuning.

**Questions To Authors:**

Gradient surgery seems to be orthogonal to the prompt pre-training stages. How does it perform when combined with the baselines like PPT and SPoT?

**Reasons To Accept:**

1. The paper is clearly written and method is easy to implement.
2. The empirical results are sufficient to verify the effectiveness of the method.

**Reasons To Reject:**

1. More parameter efficient methods like LoRA are more commonly used in current LLMs and usually achieves better performance than prompt tuning. It would be better to examine the method's effectiveness on LoRA [1].
2. Compared to the baselines (PPT [2], SPoT [3]), DawGen seems to introduce additional computation cost because it requires data generation for each downstream task while PPT and SPoT pre-train the prompt and applies it to various end tasks. This computational cost should be addressed.

[1] LoRA: Low-Rank Adaptation of Large Language Models.
[2] PPT: Pre-trained Prompt Tuning for Few-shot Learning.
[3] SPoT: Better Frozen Model Adaptation through Soft Prompt Transfer.

---

> ### Author Rebuttal · Authors · 2024-05-29
>
> We sincerely thank the reviewer for recognizing the presentation and evaluation of this work. We hope to address your concerns below.
>
> **Weakness: examine the method on other parameter-efficient learning methods.**
>
> As our framework is not limited to prompt tuning, we believe it can also improve the few-shot learning performance of LoRA. We appreciate your suggestion and will explore this further after rebuttal due to time constraints.
>
> **Weakness: the computational cost.**
>
> Comparing the computational costs of different training paradigms directly is challenging due to variations in training data, pretraining strategies, and stopping conditions. Specifically:
>
> * PPT: It involves a one-time pre-training of prompts on 10GB of OpenWebText data using next sentence prediction. While these prompts are reusable across tasks, they may underperform in tasks that are underrepresented in the OpenWebText corpora.
>
> * SPoT: Prompts are pre-trained on one or more source tasks, and then adapted for a target task. However, selecting appropriate source tasks requires an extensive search and may struggle with large domain gaps.
>
> * DawGen: It involves a one-time adaptation of the generator to the target task using few-shot examples. The adapted generator can then produce unlimited synthetic data, enhancing the flexibility of prompt training. For instance, generating 1000 examples per class only takes about two hours for paraphrase detection tasks and three hours for NLI tasks, but results in a substantial improvement on average. More advanced techniques such as FastGen [1] can further reduce the generation cost.
>
> It is unfair to compare the finetuning time for PPT and SPoT with ours, which trains from scratch. Our primary goal is to enhance few-shot performance. There may be a trade-off between performance and computational cost. Our method is a good alternative when PPT and SPoT fail in new domains.
>
> **Can Gradient surgery be combined with PPT and SPoT?**
>
> No. Gradient surgery requires access to all different domains' data in order to measure whether there is interference between the gradients of task loss w.r.t. prompt parameters from these domains. Both PPT and SPoT involve pretraining the prompts and subsequently discarding the data. We cannot obtain the gradients from the pre-training data.
>
> [1] Model Tells You What to Discard: Adaptive KV Cache Compression for LLMs

---

> > ### Comment · Reviewer_T1e9 · 2024-06-05
> > **Response to the Authors' rebuttal**
> >
> > Thanks for your response. After reading your response, I think my rating is ok and I'm not change it.

---

### Official Review · Reviewer_GDbm · 2024-05-09

**Rating:** 7
**Confidence:** 3
**Ethics Flag:** 1

**Summary:**

The paper introduces a novel approach to enhance prompt tuning (PT) in few-shot learning settings by synthesizing task-specific labeled datasets using large language models (LLMs). The authors propose a method called Distribution-Aligned Weighted Generator Tuning (DawGen) to create in-distribution synthetic data, and they apply gradient surgery to manage gradient conflicts when training with both real and synthetic data. Their experiments across various datasets demonstrate that this method can outperform traditional fine-tuning techniques, presenting an efficient alternative when large labeled datasets are not available.

**Questions To Authors:**

1.How robust is the proposed method to variations in the quality of synthetic data? Can the model's performance degrade significantly with lower-quality synthetic datasets?

2. How does the performance of DawGen scale with the size of the LLMs used for generating synthetic data? Would larger or smaller models significantly affect the outcomes?

**Reasons To Accept:**

1. The paper proposes an original method for overcoming the data limitations of few-shot PT by generating synthetic data, which is significant for advancing the efficiency of using LLMs.

2. The experiments show that their method not only competes well with but occasionally outperforms much larger models trained with conventional techniques.

3. The paper includes extensive evaluations across several datasets, demonstrating the robustness and versatility of the proposed methods.

4. The paper is well-organized, making complex concepts accessible and clearly explaining the methodology and its significance.

**Reasons To Reject:**

1. While the method shows interesting results, there is not a unique method that has the best results. The best results are scattered between different methods.

2.Some parts of the methodology, particularly around the implementation of gradient surgery and the generation of synthetic data, could be more detailed to ensure reproducibility.

3. Are the superior results statistically significant.

---

> ### Author Rebuttal · Authors · 2024-05-31
>
> We sincerely thank the reviewer for recognizing our motivation, presentation, and evaluation and providing us with constructive suggestions.
>
> **Weakness: there is not a unique method that has the best results**
>
> We recognize that no single method consistently outperforms others across all datasets. Our approach averagely performs well, but its effectiveness can vary depending on the specific dataset and task. While FT excels in some tasks, it fails in others. Considering the balance of parameter efficiency and performance, our method remains a viable option for improving prompt tuning in few-shot settings. We will add a new section of limitations to expand this discussion.
>
> **The methodology could be more detailed to enhance reproducibility**
>
> Thank you for the suggestion! We will include more details and clear examples.
>
> **Are the superior results statistically significant?**
>
> We conduct t-test for Table 1 on results where our method is superior. The table below shows the p-values. Our method shows statistically significant improvement over the second-best baseline, FT. Although not superior in every task, our method shows higher average improvements while using only 102K parameters, compared to 770M in full-model finetuning. We will include significance tests for all tables in the Appendix. Thank you for the question.
>
> Table. p-value for ours against FT (lower better).
> | model    | QQP| MRPC| MNLI| SNLI | QNLI | RTE | SICK |
> | -|-| -| - |- |- |-|- |
> | T5-large | - | 0.014 | 8e-3 | - | 8e-3 | 9e-4 | -|
> |Flan-T5-large| 0.2| 0.002 | - |  - |  9e-8 | - |  - | -|
>
> **Can the model's performance degrade significantly with lower-quality synthetic datasets?**
>
> As shown in Table 3, the performance of DawGen generally surpasses that of FewGen for both T5-large and Flan-T5-large models because DawGen produces higher quality data than FewGen due to its distribution alignment regularization.
>
> **How does the performance of DawGen scale with the size of the LLMs used for generating synthetic data?**
>
> The generator LLM, CTRL, used by FewGen and DawGen, is available only in a single size—1.63B parameters. We couldn't assess the impact of generator sizes. DawGen should theoretically work unless the LLM generates perfect synthetic data without distribution gaps. We plan to evaluate on stronger LLMs like LLaMA-13B in the future.

---

> > ### Comment · Reviewer_GDbm · 2024-06-02
> > **Response to the Authors' rebuttal**
> >
> > Thanks for your response. After reading your response, I think my rating is ok and I'm not change it.

---

### Official Review · Reviewer_1fLs · 2024-05-10

**Rating:** 4
**Confidence:** 3
**Ethics Flag:** 1

**Summary:**

This paper addresses the topic of prompt tuning, which typically requires large quantities of labelled training data and underperforms full-model fine-tuning in few-shot settings (by which the authors refer to low-resource settings rather than few-shot prompting). The authors present a method to create synthetic data using an LLM to do prompt tuning on and a method to fine-tune on the data, consisting of using gradient surgery when fine-tuning on a mix of the synthetic and real data.

Their method consists in (i) training a soft prompt using a few-shot dataset to adapt an LLM to a given domain, (ii) using their method Distribution-Aligned Weighted GENerator tuning (DawGen) to align generated examples to those of the same class, and (iii) using label-conditional prompts and (iv) using gradient surgery when fine-tuning on a combination of synthetic and real data to account for possibly low quality data.

They test their method using T5-large and FlanT5-large for 2 tasks (paraphrase detection and NLI) over 7 datasets, comparing to multiple other methods (fine-tuning, zero-shot, in-context learning, etc.). Their method gives results better than fine-tuning for some tasks and worse for others, and 18% improvement on average compared to basic prompt tuning for T5-large.

In its current state, I found the paper unclear with respect to its main aim (more with respect to the writing of the paper - providing motivation for working with so few training examples for example) and would also have appreciated some more details concerning the method (see below). The paper is also lacking examples allowing me to review the quality of the produced synthetic data.

**Questions To Authors:**

1. The meaning of “few-shot learning” was initially unclear to me (between using few-shot examples and using a very small set of  training examples), particularly as the article is on the topic of LLMs, which typically use few-shot examples. I assume that the article wants to deal with very few training examples, but I only got this after rereading several times, and putting the article in the context of (Gao et al., 2022).
2. Synthetic data generation method:

    a) It is Unclear to me what the input to the LLM is other than the label-conditioned soft prompt and the current example.

     b) What is the quality of the generated data like? There are currently no examples of generated examples in the article, making it impossible for me as a reviewer to judge the performance of the generation method. This is particularly relevant as the authors cite the inevitability of low quality data.
3. Missing details:

    a) Why was so little data used? 16 examples for each class? Is this a realistic setting? What about other amounts of data?

    b) Amount of synthetic data generated. How diverse is the data? Are there duplicates?

    c) In-context learning -> how many examples were used?

    d) What does the “number params” correspond to in Table 1? The models are the same, so does this refer to the number of tuned parameters?

    e) PPT appears to use T5-xxl and not T5. If so these results are not comparable.

4. In the related work, the authors mention that fine-tuning on a model on few examples (e.g. 16 samples per class) can lead to overfitting. What is the significance of 16 samples? i.e. why choose this number, which appears to me to be far too low to be realistic.
5. Verbaliser -> it would be good to make it clear what you mean by this in the heart of the text (it only became clear to me from the Appendix)
6. Very low scores of the baselines “Prompting” and “In-context”, especially given the binary nature of the tasks. Why do they do so poorly (Under random for some tasks)? Could this mean that the base model being used is not good enough and perhaps a better model should be used?

## Language, grammar, etc.

- Few-shot real data -> real few-shot data
- Comparable performance as -> comparable performance to
- E.g., -> it is good practice to use a non-breaking space after E.g. to avoid linebreaks
- Pertaining data -> Pretraining data
- Which is inevitable to contain low-quality data -> which inevitably results in low quality data
- Lester et al. (2021) converts -> Lester et al. (2021) convert
- Taking sentence-pair classification as an example. Given -> Taking sentence-pair classification as an example, given

**Reasons To Accept:**

- It is interesting to have a comparison of different approaches to using LLMs and I appreciate the effort in combining methods to use the data in the best way

**Reasons To Reject:**

- Lack of general clarity in the paper about the main aim of the article. This is in my opinion linked to several things, starting from an unclear motivation concerning the use of “few-shot”, which in the context of prompting often refers to in-context learning, but here appears to refer to the case where we have few training examples (this could be made much clearer and non-ambiguous right from the start). On first reading, I was not sure why the authors were only using 16 examples per class of real data, which is so few. Also, why only choose 16 examples? If the main aim of the paper is to study whether prompt tuning can rival fine-tuning in low-resource settings, I think it would have been more informative to have the comparison at different numbers of training data examples in order to see the progression of the 2 approaches. With so few examples, fine-tuning must be hugely overfitting too. Is this a realistic setting?
- In the main method, it was unclear to me what the input to the LLM looks like in order to produce synthetic training examples. Does it just consist of the label-conditional soft prompt and one of the 16 examples (for each class)? Also, I may have missed it, but how many examples were generated?
- A major weakness in my opinion is the fact that no examples of generation data or predictions are given in the example. As a reviewer, this makes it impossible for me to judge the performance of the generation method and also whether the base LLM used is up to the job at all. The very low accuracy scores of the baseline on the binary tasks (below random) appear to show that the base model used may be inadequate.

---

> ### Author Rebuttal · Authors · 2024-05-30
>
> Thank you for the comments. I would like to clarify two terms. Prompt tuning is an efficient, low-cost way of adapting an LLM to new downstream tasks without retraining the model and updating its weights; Few-shot learning aims to make accurate predictions by training on a very small number of labelled examples. These concepts are distinct from providing few shots in prompting ChatGPT (in-context learning). We will enhance the description in our revised manuscript.
>
> **Weakness: Lack of clarity about the aim**
>
> While prompt tuning is efficient, it often lags behind traditional FT when training data is scarce. Hence, we propose synthesizing data for augmentation. As synthetic data may not be of high quality, we propose the gradient surgery to enhance learning, as stated in abstract and introduction.
>
> Few-shot is a realistic setting, especially in rare or dynamic situations. Prior research choose 8 or 16 training samples (shots) as a standard routine. We follow LM-BFF,PPT, and OPTIMA to set the shots to 16 for fair comparison.
>
> For a comparison of FT and PT under the progression of training data, see Figure 2 in the PPT paper and Figure 3 in the OPTIMA paper.
>
> **Weakness: the base model used may be inadequate**
>
> Low accuracy scores are due to 1) unbalanced test sets of the binary tasks; and 2) challenging tasks. Even if we use a more powerful base model (T5-xxl, 11B), the accuracy is not sufficiently good.
>
> Table. Reported Test acc on RTE.
> | Paper    | Base Model | Strategy | Accuracy |
> | - | - | - |-|
> | LM-BFF | RoBERTa_large | Prompting | 51.3 |
> | | | FT |54.4 |
> | PPT |   T5-xxl |  PT |  53.5 |
> | Ours    |   T5-large | Prompting |  52.35 |
> | | |PT |  51.70 |
> | | | FT | 55.81|
>
>
> **Question: Synthetic Examples**
>
> The inputs for our generator are "soft prompt + hard prompt + label (+premise for NLI)". We generate 1000 synthetic samples per class. Data from FewGen is available at: https://github.com/yumeng5/FewGen/tree/master/data/k-shot. We put our examples and more analysis here: https://drive.google.com/file/d/1yWj5E6mvN5oIQbWhobC29taEEGKXtox7/view?usp=sharing.
>
> **Other questions**
>
> *Num. Params* => #trainable Params
>
> *PPT with T5-xxl* lags behind our method with a weaker model, T5-large. The conclusion still holds.
>
> *ICL* => randomly select 8 examples from the few-shot set, followed by an unlabeled test example for prediction. Run 5 random seeds and average test performance.

---

> > ### Comment · Reviewer_1fLs · 2024-06-05
> >
> > Thank you for your detailed response. I do believe that the aim/motivation of the paper could be made clearer with some reformulations in the text, along the lines of what you write in your response. I think that the examples of generated synthetic data - I think this would be a useful contribution to the supplementary material, although it would have been good to have a more detailed analysis of the generated data.
> >
> > I maintain that it would be more interesting to test different numbers of training examples other than 16 per class, which is very few (even if this is what some other works have done, I am not convinced that it that realistic). In reality, with a bit of annotation it is often possible to have more examples than 16 per class. I would therefore find it realistic to test the method with varying amounts of data to ascertain what the threshold is for prompt tuning still lagging behind FT.
> >
> > I therefore choose to keep the same score.

---

> > > ### Author Response · Authors · 2024-06-06
> > > **Regarding few-shot learning and increasing the amount of shots**
> > >
> > > I want to clarify that **“with a bit of annotation …”** is impossible for many real-world tasks especially for rare diseases. The assumption of the availabity of sufficient labeled data is unrealistic. Many machine learning fields have been dedicated to solve this problem including transfer learning, zero-shot and few-shot learning (FSL). FSL focuses on enabling models to learn new tasks from a very limited amount of data – typically only a few examples, hence the term "few-shot". I won’t repeat the facts and significance about FSL but refer you to the survey [1].
> > >
> > > * Scale is the power of today’s LLMs! Preceding research established many neural scaling laws for deep learning models [2,3]. It is a known fact that as the amount of training data decreases, the model’s performance degrades significantly. If annotated data is not a problem, maybe one panacea can cure all diseases. Unfortunately, data is a big problem! **Few-shot setting is at the tail of the scaling law** and **abundant research have been done to elevate the performance of this extreme case at the tail**. These methods predominantly leverage external knowledge, such as pretraining and transfer learning, to guide the model in learning effectively from a few examples.
> > >
> > > Based on these findings and limitations, we **focus on pushing the boundary of prompt tuning in few-shot settings with the help of generative AI.**
> > >
> > > * Studying scaling laws involves non-trivial computational costs. This is not the focus of this paper (as we presented many other interesting findings if you read the paper), therefore we referred you to previous studies conducted **exactly for comparing prompt tuning and full-model tuning given different amounts of real data**. Scaling up in our problem setting is **extremely resource-intensive** and deviates from our main objective. *It involves generator tuning, data generation, and prompt tuning for every random seed and every hyperparameter setting*. For example, suppose we fix 5 random seeds and 3 hyperprameter settings, and want to plot 10 points for drawing the scaling law for a single NLI task, it would approximately take about (2h + 3h + 2h) * 5 * 3 * 10 = 1050 hrs = 43 days on the A6000 workstation for both training and evaluation. This study obviously does not worth the costs compared to other more important studies. Please allow this paper to present limited (but nontrivial) findings as we can only push forward the science step by step.
> > >
> > > The opportunity for us to conduct rebuttal and discussion is to solve your questions to avoid biased scores, encourage insightful discussions and for us to receive constructive suggestions to improve the paper. It is not meant for debating on some already established facts and problems, for which we should remain honest and scientific. Since I have addressed most of your questions, I would expect you to justify your score with sound reasons and facts, presented professionally. I really hope you can spend a little more time to reread and understand this paper's contributions, and also want to refer you to other reviewers’ justifications. We’re actively working on improving the paper and hope to receive your constructive suggestions and more importantly, a fairer score!
> > >
> > >
> > > ## A take-away summary of the key contributions
> > >
> > > Although it has been clearly stated at the beginning of this paper, we want to clarify in short what our contributions are for your reference.
> > >
> > > Prompt tuning, conditioned on LLMs, has demonstrated competitive performance as full-model tuning under full-data settings. However, recent studies uncovered that prompt tuning significantly underpeforms full-model tuning in few-shot settings, limiting its application scope (Problem). Conventional methods generally rely on transfer learning from real-world datasets, which assumes the availability of large-scale, real-world, task-relevant datasets (Limitations of existing solutions). This paper builds on recent research for “LLMs as training data generators” to provide an alternative solution for advancing prompt tuning in few-shot settings (Motivation). We identified the problem of potential ill distributions in synthetic datasets and proposed DawGen to promote distribution alignment for synthetic data generation (Contribution 1). We want to make the best use of few-shot and synthetic data to further enhance downstream learning performance. Hence, we employ gradient surgery to modify the gradients from synthetic data by always projecting them onto the direction of real-data gradients and subtracting conflicting components (Contribution 2).
> > >
> > >
> > > References
> > >
> > > [1] A Comprehensive Survey of Few-shot Learning: Evolution, Applications, Challenges, and Opportunities
> > >
> > > [2] Beyond neural scaling laws: beating power law scaling via data pruning. NeurIPS 2022
> > >
> > > [3] Revisiting Neural Scaling Laws in Language and Vision.  NeurIPS 2022.
> > >
> > > [4] Few-Shot Parameter-Efficient Fine-Tuning is Better and Cheaper than In-Context Learning. NeurIPS 2022.

---

> ### Comment · Area_Chair_Akiy · 2024-06-05
> **Last Day of Discussion**
>
> Dear Reviewer 1fLs,
>
> Thanks for your review! The discussion is ending tomorrow, and it'd be greatly appreciated if you could acknowledge the author's rebuttal and update your review if necessary.
>
> Thank you!
> AC

---

### Comment · Area_Chair_Akiy · 2024-06-04
**Author-Reviewer Discussions**

Dear reviewers,

Thank you for your review of this submission. The authors have provided their rebuttal, and it is time to engage in discussions with them if you have additional questions. Please take the time to read the other reviews and update your own review as necessary.

Reviewers 1fLs/T1e9: Even if you do not have further questions and/or choose to maintain your original score, please acknowledge that you have read the authors' rebuttal.

Thank you!
AC

---

### Decision · Program_Chairs · 2024-07-10

**Decision:**

Accept

**Comment:**

The paper studies few-shot prompt tuning enhanced with data augmentation. The general idea is to use LLMs to generate synthetic data for training the soft prompts. The proposed method, Distribution-Aligned Weighted Generator Tuning (DawGen), aims to produce in-distribution synthetic data that are aligned with few-shot samples. The training procedure consists of a gradient surgery method to mitigate gradient conflicts when training the model with both real and synthetic data. The reviewers generally believe that the paper is well-motivated and clearly-written, and find the proposed method to be novel and effective.

While the AC believes that the paper has enough merits to warrant acceptance, some of the authors' responses appear unprofessional and disrespectful toward the reviewers. One of the reviewers expressed concerns about the tone and content of the authors' response. It is natural that authors/reviewers have different opinions, but please note that the reviewers have dedicated their time and efforts to reviewing the manuscript, and it is crucial that all interactions remain respectful and professional.

[At least one review was discounted during the decision process due to quality]